# Distinguishing Seed Cultivars of Quince (*Cydonia oblonga* Mill.) Using Models Based on Image Textures Built Using Traditional Machine Learning Algorithms

**Ewa Ropelewska [1,\*], Dorota E. Kruczyńska [2] and Monika Mieszczakowska-Frąc [1]**

[1] Fruit and Vegetable Storage and Processing Department, The National Institute of Horticultural Research, Konstytucji 3 Maja 1/3, 96-100 Skierniewice, Poland; monika.frac@inhort.pl

[2] Cultivar Testing, Nursery and Gene Bank Resources Department, The National Institute of Horticultural Research, Konstytucji 3 Maja 1/3, 96-100 Skierniewice, Poland; dorota.kruczynska@inhort.pl

\* Correspondence: ewa.ropelewska@inhort.pl

**Abstract:** Different cultivars of seeds may have different properties. Therefore, distinguishing cultivars may be important for seed processing and product quality. This study was aimed at revealing the usefulness of innovative models developed based on selected image textures built using traditional machine algorithms for cultivar classification of quince seeds. The quince seeds belonging to four cultivars 'Uspiech', 'Leskovac', 'Bereczki', and 'Kaszczenko' were considered. In total, 1629 image textures from different color channels for each seed were extracted from color images acquired using a flatbed scanner. Texture parameters were used to build models for a combined set of selected textures from all color channels, sets of selected textures from color spaces RGB, Lab, and XYZ, and individual color channels *R*, *G*, *B*, *L*, *a*, *b*, *X*, *Y*, and *Z* using algorithms from different groups. The most successful models were developed using the Logistic (group of Functions), IBk (Lazy), LogitBoost (Meta), LMT (Trees), and naïve Bayes (Bayes). The classification accuracy reached 98.75% in the case of a model based on a combined set of textures selected from images in all color channels developed using the Logistic algorithm. For most models, the greatest misclassification of cases was observed between seeds 'Bereczki' and 'Kaszczenko'. The developed procedure can be used in practice to distinguish quince seeds in terms of a cultivar and avoid mixing seed cultivars with different properties intended for further processing.

**Keywords:** quince seeds; color imaging; image processing; texture parameters; cultivar classification

## 1. Introduction

Quince (*Cydonia oblonga* Mill.) is a plant belonging to the Rosaceae family. It is widely cultivated in Europe, Asia, and the Middle East [1]. Quince is native to Trans-Caucasia and northern Iran and has spread to other regions, such as Europe and America [2,3]. In Europe, quince is cultivated mainly in central and southern regions with a higher temperature in summer. Although quince is self-pollinated, pollination with other trees can result in improving the yield. Quince rootstocks can be applied in the case of pear trees to increase the yield, exert a dwarfing effect on the trees and suppress excessive vegetative growth [4].

Quince fruit is a pome containing numerous seeds. The fruit is big with asymmetric shapes, variable dimensions, and a characteristic fragrance. An abundant hair covering the peel disappears with fruit ripening. The quince flesh is white-yellow and firm and easily oxidized when exposed to air [5]. The ripe quince fruit belonging to selected cultivars has a pleasant, powerful, and lasting flavor [6]. However, despite the pleasant and intense aroma, most cultivars are commonly unacceptable to be eaten raw due to

hard, astringent, and sour flesh. Therefore, quince is commonly cooked or processed into jam, jelly, marmalade, puree, pudding, dried slices, juice, compote, wine, or liqueur [1,2,7,8]. Quince is mainly a traditional economic resource in family farms and small producers [9]. Because of its nutritional value, great health benefits, and limited use at home and in industries, quince is recognized as a target for future investment and emphasis [7].

The edible part of the quince fruit is characterized by its health-promoting properties. Quince is a source of antioxidants and is characterized by its pharmaceutical, nutraceutical, and ornamental properties [1]. The flesh is composed of organic acids, sugars, and polysaccharides, as well as proteins, phenolics, vitamins, and lipids. Especially, a high content of phenolics is present in raw quince [3]. Due to the presence of functional compounds and phytochemicals, it can be used in the prevention and treatment of diseases [7]. Due to the presence of biologically active compounds, such as polyphenols, vitamin C, and terpenoids, quince fruit can help treat sore throat, bronchitis, and constipation. Quince leaves have constituents, which can be effective against diabetes, cancer, or hyperlipidemia. Furthermore, quince seeds, due to their tannin contents, can have a strong anti-diarrheal activity [1].

It was found that the quince seed mucilage is very valuable. Among others, it is used in the food industry as a thickener and bulking agent in several products [7]. Quince seed mucilage is also used as a biopolymer for the pharmaceutical industry. It includes a mixture of cellulose and water-soluble polysaccharide. Furthermore, acidic hydrolysis revealed the presence of D-xylose, haloboronic acids, and L-arabinose. Quince seed mucilage is used as an adjuvant in the manufacturing of pharmaceutical products and is characterized by binding, stabilizing, thickening, disintegrating, humidifying, suspending, sustaining, and emulsifying properties, at different proportions in various pharmaceutical dosage forms [10].

The properties of seeds can vary among cultivars [11,12]. Therefore, the correct distinguishing and identification of cultivars can be important before seed processing. Seed quality can be evaluated by trained experts and other techniques, including seed image analysis, which is useful for preserving biodiversity. Image analysis techniques have advantages as they can speed up the process and automatically classify the seed features based on the image pixel values [13]. Image processing allows for achieving accurate and rapid results. The seed classification provides important information on seed quality. The identification of seed cultivars by human base perception can be a difficult task. The lack of human resources is also a problem. Technology using seed images helps to evaluate cultivars. The seed cultivar identification by analyzing digital images through machine learning techniques overcomes the issues related to human visual perception [14]. Automatic methods of seed classification are more effective than the manual process, which is more difficult and time-consuming, especially at high production volumes [15]. A machine vision system can be an alternative to the human inspection of seed cultivars to classify them in terms of their quality [16]. The imaging techniques combined with machine learning algorithms reveal good results in seed classification [17].

Image processing techniques can be considered the foundation of seed classification. The classification of seeds by cultivar based on image features using machine learning techniques can be successively used in research [18]. Computer vision studies mainly concentrate on recognizing and obtaining features from images using texture, color, and morphological features [19]. For seed identification and classification, seed texture, color, size, shape, and spectral reflectance are important phenotypic features. The key steps of seed identification using machine vision are feature extraction and classification. Feature extraction quantitatively describes morphological, color, and textural features by empirical formulas and the various algorithms allow for seed classification with different performances. The acquiring features non-destructively using machine vision and the development of automatic algorithms results in the possibility of accurate and rapid seed cultivar classification [20].

Despite promising reports on the possibility of using image analysis and machine learning in seed cultivar classification, there is a lack of information on the comprehensive use of this approach for quince seeds. The objective of this study was to determine the usefulness of an approach combining image processing and traditional machine learning to classify quince seeds in terms of cultivars. In total, 1629 image texture parameters (181 for each of the nine color channels) for each seed were computed. The novelty of the study involved developing models to distinguish quince seed cultivars based on different textures selected from a combined set of textures selected from all color channels, individual color spaces RGB, Lab, and XYZ, and individual color channels *R*, *G*, *B*, *L*, *a*, *b*, *X*, *Y*, and *Z*. Models were built using algorithms from different groups of Functions, Lazy, Meta, Trees, and Bayes. Thus, the present study is an innovative and comprehensive approach to the cultivar classification of quince seeds.

## 2. Materials and Methods

### 2.1. Materials

The seeds belonging to four quince cultivars 'Uspiech', 'Leskovac', 'Bereczki', and 'Kaszczenko' were used in the experiment. The quinces were collected from the Experimental Orchard of the National Institute of Horticultural Research in Dąbrowice (Poland). The sampled quinces were at harvest maturity and were stored at a room temperature of 20 ± 1 °C for a month. In the case of each cultivar, fifteen fruits were used. The whole quinces were cut using a sharp knife and seeds were manually extracted from the fruit. The seeds were rinsed under tap water and cleaned. This procedure lasted a few minutes for each seed cultivar and then seeds were immediately subjected to image acquisition.

### 2.2. Image Acquisition and Processing

The quince seed images were acquired using an Epson Perfection V19 flatbed scanner (Epson, Suwa, Nagano, Japan). The scanner was placed in a box with black walls inside. Thus, the acquired images were on a black background that facilitated the image segmentation. Before scanning, color calibration of the scanner was carried out. The color images of quince seeds were acquired at a resolution of 800 dpi and saved in TIFF format. For each quince cultivar 'Uspiech', 'Leskovac', 'Bereczki', and 'Kaszczenko', images of one hundred seeds were obtained and used for image processing. Twenty-five seeds were in each image. In the beginning, the file format of images was changed to BMP. The images were processed with the use of the MaZda software (Łódź University of Technology, Łódź, Poland) [21–23]. The quince seed images conversion to color channels *R*, *G*, *B*, *L*, *a*, *b*, *X*, *Y*, and *Z* was performed. The original seed images and images in selected color channels are presented in Figure 1. The image segmentation was performed using a brightness threshold. The images were segmented into lighter quince seeds and a black background. Each seed was separated from the background and considered as one region of interest (ROI). Then, 1629 image texture parameters were computed for each ROI, including 181 textures for each color channel based on the histogram, gradient map, Haar wavelet transform, autoregressive model, co-occurrence matrix, and run-length matrix.

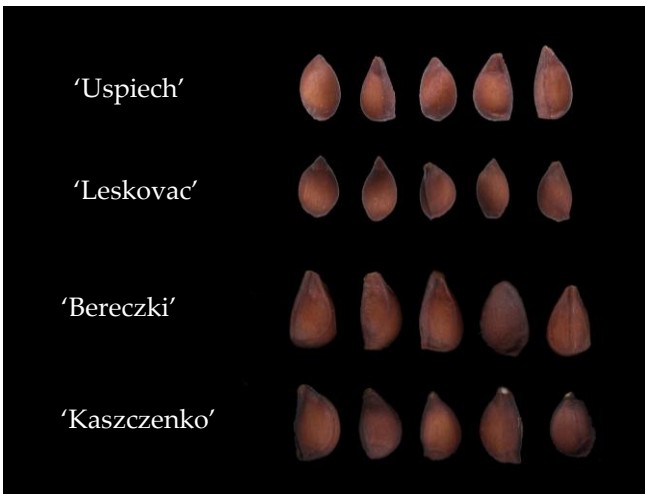

**Figure 1.** The exemplary original images of quince seeds belonging to different cultivars.

*2.3. Cultivar Classification of Quince Seeds Based on Image Texture Parameters*

The differences in selected texture parameters and image texture between quince seed cultivars were determined at a significance level of $p < 0.05$ using STATISTICA 13.1 (Dell Inc., Tulsa, OK, USA, StatSoft Polska, Kraków, Poland). The normality of the distribution of variables and homogeneity of variance was checked. Then, a Newman–Keuls test was used to compare the means.

The quince seed cultivars were classified using the WEKA machine learning software (Machine Learning Group, University of Waikato, Hamilton, New Zealand) [24–26]. The classification models were built based on selected image textures to distinguish seeds belonging to 'Uspiech', 'Leskovac', 'Bereczki', and 'Kaszczenko'. The attribute selection was carried out for a set of textures extracted from images in all color channels, and sets of textures from individual color spaces RGB, Lab, and XYZ, and color channels *R*, *G*, *B*, *L*, *a*, *b*, *X*, *Y*, and *Z* using the Best First and Correlation-based Feature Selection subset evaluator. The models were developed using machine learning algorithms from the groups of Functions, Lazy, Meta, Trees, and Bayes using a 10-fold cross-validation. The confusion matrices, overall accuracies, and the values of the TP (True Positive) Rate = Recall, FP (False Positive) Rate, Precision, PRC (Precision-Recall) Area, ROC (Receiver Operating Characteristic) Area, MCC (Matthews Correlation Coefficient), and F-Measure were computed [27–29].

## 3. Results and Discussion

The confusion matrices and overall accuracies for models developed for a combined set of textures selected from all color channels *R*, *G*, *B*, *L*, *a*, *b*, *X*, *Y*, and *Z* of seed images are presented in Figure 2. The combined textures of the images in each of the channels were useful for building the classification model. The textures with the highest power to distinguish quince seed cultivars were, among others, RSGNonZeros, RSGArea, GHMean, GHDomn10, BHMean, BHVariance, LHMean, LS5SN5SumAverg, aHMean, aHPerc99, bSGNonZeros, bS4RHGLevNonU, XS5SN1SumOfSqs, XS4RVGLevNonU, YS5SH1SumAverg, ZHMean, ZHMaxm10. The mean comparison of selected image textures is presented in Table 1.

**Table 1.** The selected texture parameters of quince seeds.

| Class | RSGNonZeros | GHMean | BHMean | LHMean | aHMean | ZHMean | ZHMaxm10 |
|---|---|---|---|---|---|---|---|
| 'Uspiech' | 0.992 a | 74.22 a | 63.27 a | 116.32 a | 142.47 a | 15.78 a | 0.678 a |
| 'Leskovac' | 0.988 b | 60.21 b | 53.83 b | 99.96 b | 139.06 b | 10.27 b | 0.928 b |
| 'Bereczki' | 0.987 c | 45.55 c | 39.86 c | 84.36 c | 137.82 c | 6.07 c | 0.956 c |
| 'Kaszczenko' | 0.990 d | 47.45 d | 42.96 d | 85.77 c | 137.37 c | 6.58 c | 0.985 d |

a,b,c,d—the same letters in the columns denote no statistical differences.

The seeds of 'Uspiech', 'Leskovac', 'Bereczki', and 'Kaszczenko' were correctly classified with an overall accuracy reaching 98.75% for the Logistic algorithm (Figure 2a). It was observed that seeds belonging to quince 'Bereczki' (class 3) were completely correctly classified with an accuracy equal to 100%. Whereas accuracies for 'Uspiech' (class 1) and 'Leskovac' (class 2) reached 99% and for 'Kaszczenko' (class 4)—97%.

The lowest overall accuracy (97.50%) and the greatest mixing of cases were observed for the model built using naïve Bayes (Figure 2e). For seeds of individual quince cultivars, the accuracies of 100% for 'Leskovac', 99% for 'Uspiech', 98% for 'Bereczki', and 93% for 'Kaszczenko' were determined.

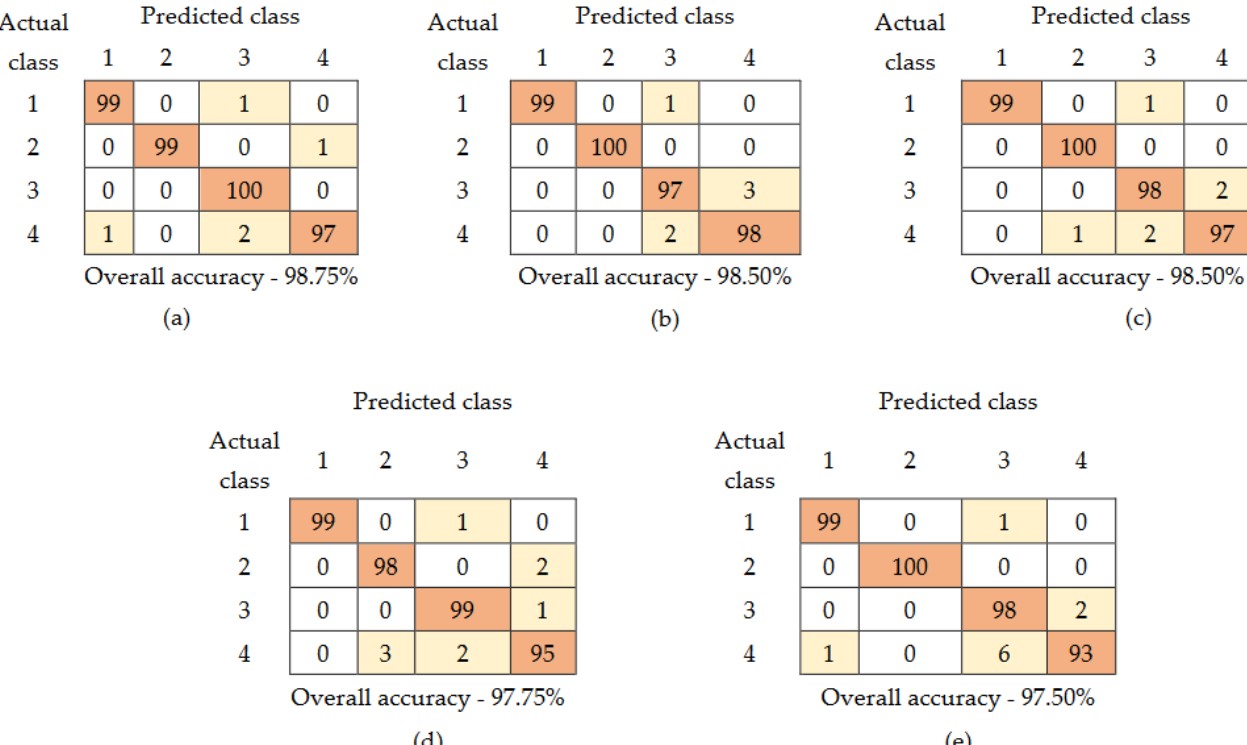

**Figure 2.** The confusion matrices of distinguishing quince seed of 'Uspiech' (class 1), 'Leskovac' (class 2), 'Bereczki' (class 3), and 'Kaszczenko' (class 4) using models based on combined selected image textures from color channels *R, G, B, L, a, b, X, Y,* and *Z* developed using Logistic (**a**), IBk (**b**), LogitBoost (**c**), LMT (**d**), naïve Bayes (**e**). Orange color—correctly classified cases, yellow color—incorrectly classified cases.

In the case of models built using a combined set of textures selected from images in color channels *R, G, B, L, a, b, X, Y,* and *Z*, the performance metrics, such as TP (True Positive) Rate, FP (False Positive) Rate, Precision, PRC (Precision-Recall) Area, ROC (Receiver Operating Characteristic) Area, MCC (Matthews Correlation Coefficient), and F-Measure were very high for each machine learning algorithm (Table 2). The highest TP Rate equal to 1.000 was determined for seeds of 'Bereczki' for Logistic and 'Leskovac' for

IBk, LogitBoost, and naïve Bayes. It meant that these cultivars were classified with an accuracy of 100%. The lowest TP Rate of 0.930 was found for seeds 'Kaszczenko' in the case of naïveBayes and it meant that cultivar 'Kaszczenko' was classified with the lowest correctness for a model built using naïveBayes. The values of FP Rate ranged from 0.000 for 'Leskovac' and Logistic, IBk, and naïveBayes and 'Uspiech' for IBk, LogitBoost, and LMT to 0.023 for 'Bereczki' and naïve Bayes. An FP rate equal to 0.000 for a given class indicated that no case from other classes was included in that class. The highest FP Rate meant that most cases belonging to other cultivars were classified as the cultivar with the highest value of this performance metric. The highest value of Precision reached 1.000 for 'Leskovac' (Logistic, IBk, naïveBayes) and 'Uspiech' (IBk, LogitBoost, LMT). The other metrics, PRC Area, ROC Area, MCC, and F-Measure were equal to 1.000 in the case of quince seeds 'Leskovac' for models built using IBk and naïveBayes. The values of PRC Area and ROC Area also reached 1.000 for seeds 'Leskovac' and LogitBoost algorithm.

**Table 2.** The performance metrics of quince seed classification using models built based on combined selected textures from images in color channels *R*, *G*, *B*, *L*, *a*, *b*, *X*, *Y*, and *Z*.

| Algorithm | Class | TP Rate | FP Rate | Precision | PRC Area | ROC Area | MCC | F-Measure |
|---|---|---|---|---|---|---|---|---|
| Logistic (Functions) | 'Uspiech' | 0.990 | 0.003 | 0.990 | 0.989 | 0.998 | 0.987 | 0.990 |
| | 'Leskovac' | 0.990 | 0.000 | 1.000 | 0.992 | 0.990 | 0.993 | 0.995 |
| | 'Bereczki' | 1.000 | 0.010 | 0.971 | 0.989 | 0.998 | 0.980 | 0.985 |
| | 'Kaszczenko' | 0.970 | 0.003 | 0.990 | 0.951 | 0.987 | 0.973 | 0.980 |
| IBk (Lazy) | 'Uspiech' | 0.990 | 0.000 | 1.000 | 0.992 | 0.990 | 0.993 | 0.995 |
| | 'Leskovac' | 1.000 | 0.000 | 1.000 | 1.000 | 1.000 | 1.000 | 1.000 |
| | 'Bereczki' | 0.970 | 0.010 | 0.970 | 0.936 | 0.978 | 0.960 | 0.970 |
| | 'Kaszczenko' | 0.980 | 0.010 | 0.970 | 0.956 | 0.981 | 0.967 | 0.975 |
| LogitBoost (Meta) | 'Uspiech' | 0.990 | 0.000 | 1.000 | 0.994 | 0.995 | 0.993 | 0.995 |
| | 'Leskovac' | 1.000 | 0.003 | 0.990 | 1.000 | 1.000 | 0.993 | 0.995 |
| | 'Bereczki' | 0.980 | 0.010 | 0.970 | 0.996 | 0.999 | 0.967 | 0.975 |
| | 'Kaszczenko' | 0.970 | 0.007 | 0.980 | 0.998 | 0.999 | 0.967 | 0.975 |
| LMT (Trees) | 'Uspiech' | 0.990 | 0.000 | 1.000 | 0.994 | 0.995 | 0.993 | 0.995 |
| | 'Leskovac' | 0.980 | 0.010 | 0.970 | 0.992 | 0.990 | 0.967 | 0.975 |
| | 'Bereczki' | 0.990 | 0.010 | 0.971 | 0.986 | 0.997 | 0.974 | 0.980 |
| | 'Kaszczenko' | 0.950 | 0.010 | 0.969 | 0.951 | 0.993 | 0.946 | 0.960 |
| Naïve Bayes (Bayes) | 'Uspiech' | 0.990 | 0.003 | 0.990 | 0.994 | 0.995 | 0.987 | 0.990 |
| | 'Leskovac' | 1.000 | 0.000 | 1.000 | 1.000 | 1.000 | 1.000 | 1.000 |
| | 'Bereczki' | 0.980 | 0.023 | 0.933 | 0.946 | 0.987 | 0.941 | 0.956 |
| | 'Kaszczenko' | 0.930 | 0.007 | 0.979 | 0.964 | 0.987 | 0.940 | 0.954 |

TP Rate—True Positive Rate, FP Rate—False Positive Rate, PRC Area—Precision-Recall Area, ROC Area—Receiver Operating Characteristic Area, MCC—Matthews Correlation Coefficient.

The classification models built based on selected image textures from the color space Lab produced very high overall accuracies from 95.25% (naïve Bayes) to 98.25% (LogitBoost) (Figure 3). In the case of individual quince cultivars, only seeds 'Leskovac' were completely correctly classified with an accuracy of 100% for models built using IBk and LogitBoost. Whereas seeds 'Bereczki' and 'Kaszczenko' were classified with the lowest accuracies of 91 and 92%, respectively, in the case of a model developed using naïve Bayes. The most cases of confusion were found in the classes 'Bereczki' and 'Kaszczenko'.

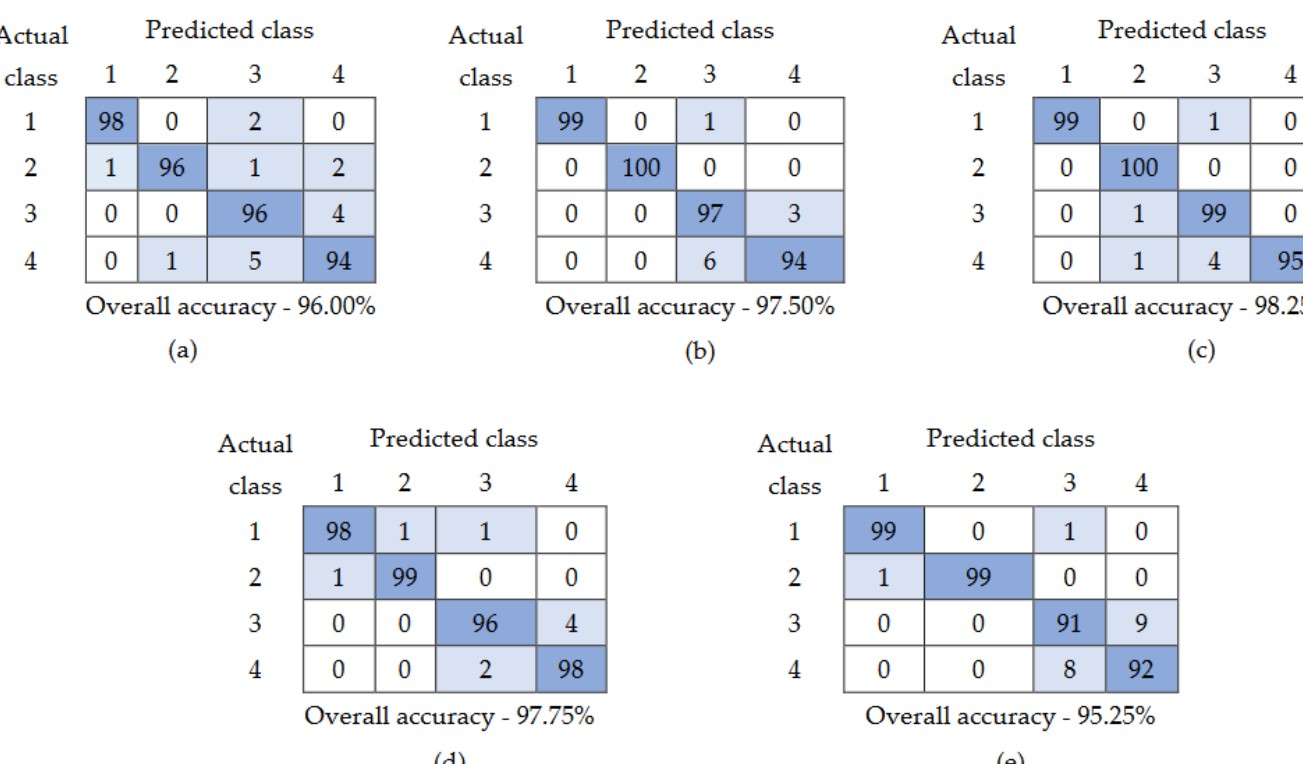

**Figure 3.** The confusion matrices of distinguishing quince seed cultivars 'Uspiech' (class 1), 'Leskovac' (class 2), 'Bereczki' (class 3), and 'Kaszczenko' (class 4) using models based on selected image textures from color space Lab developed using Logistic (**a**), IBk (**b**), LogitBoost (**c**), LMT (**d**), and naïve Bayes (**e**). Dark blue color—correctly classified cases, light blue color—incorrectly classified cases.

The performance metrics presented in Table 3 confirmed the high correctness of the classification of seeds 'Leskovac'. In the case of a model built by the IBk algorithm, the values of TP Rate, Precision, PRC Area, ROC Area, MCC, and F-Measure reached 1.000, and FP Rate was equal to 0.000. For the mentioned cultivar, the values of 1.000 were also found for TP Rate, PRC Area, and ROC Area for a model built using LogitBoost, and for Precision for a model developed using naïve Bayes. Precision equal to 1.000 and FP Rate of 0.000 were also observed for the seeds of 'Uspiech' for models built using IBk and LogitBoost and 'Kaszczenko' for LogitBoost. However, in the case of most models, the highest values of FP Rate and lowest other classification performance metrics were obtained for seeds 'Bereczki' and 'Kaszczenko'.

**Table 3.** The results of the classification of quince seeds of 'Uspiech', 'Leskovac', 'Bereczki', and 'Kaszczenko' based on models developed using selected image textures from color space Lab.

| Algorithm | Class | TP Rate | FP Rate | Precision | PRC Area | ROC Area | MCC | F-Measure |
|---|---|---|---|---|---|---|---|---|
| Logistic (Functions) | 'Uspiech' | 0.980 | 0.003 | 0.990 | 0.991 | 0.994 | 0.980 | 0.985 |
| | 'Leskovac' | 0.960 | 0.003 | 0.990 | 0.995 | 0.998 | 0.967 | 0.975 |
| | 'Bereczki' | 0.960 | 0.027 | 0.923 | 0.948 | 0.989 | 0.921 | 0.941 |
| | 'Kaszczenko' | 0.940 | 0.020 | 0.940 | 0.965 | 0.975 | 0.920 | 0.940 |
| IBk (Lazy) | 'Uspiech' | 0.990 | 0.000 | 1.000 | 0.993 | 0.996 | 0.993 | 0.995 |
| | 'Leskovac' | 1.000 | 0.000 | 1.000 | 1.000 | 1.000 | 1.000 | 1.000 |
| | 'Bereczki' | 0.970 | 0.023 | 0.933 | 0.880 | 0.966 | 0.935 | 0.951 |
| | 'Kaszczenko' | 0.940 | 0.010 | 0.969 | 0.926 | 0.938 | 0.940 | 0.954 |

| | | TP Rate | FP Rate | | | | | |
|---|---|---|---|---|---|---|---|---|
| LogitBoost (Meta) | 'Uspiech' | 0.990 | 0.000 | 1.000 | 0.994 | 0.995 | 0.993 | 0.995 |
| | 'Leskovac' | 1.000 | 0.007 | 0.980 | 1.000 | 1.000 | 0.987 | 0.990 |
| | 'Bereczki' | 0.990 | 0.017 | 0.952 | 0.993 | 0.998 | 0.961 | 0.971 |
| | 'Kaszczenko' | 0.950 | 0.000 | 1.000 | 0.994 | 0.998 | 0.967 | 0.974 |
| LMT (Trees) | 'Uspiech' | 0.980 | 0.003 | 0.990 | 0.982 | 0.991 | 0.980 | 0.985 |
| | 'Leskovac' | 0.990 | 0.003 | 0.990 | 0.993 | 0.993 | 0.987 | 0.990 |
| | 'Bereczki' | 0.960 | 0.010 | 0.970 | 0.975 | 0.996 | 0.953 | 0.965 |
| | 'Kaszczenko' | 0.980 | 0.013 | 0.961 | 0.990 | 0.993 | 0.960 | 0.970 |
| Naïve Bayes (Bayes) | 'Uspiech' | 0.990 | 0.003 | 0.990 | 0.993 | 0.991 | 0.987 | 0.990 |
| | 'Leskovac' | 0.990 | 0.000 | 1.000 | 0.995 | 0.996 | 0.993 | 0.995 |
| | 'Bereczki' | 0.910 | 0.030 | 0.910 | 0.903 | 0.987 | 0.880 | 0.910 |
| | 'Kaszczenko' | 0.920 | 0.030 | 0.911 | 0.978 | 0.984 | 0.887 | 0.915 |

TP Rate—True Positive Rate, FP Rate—False Positive Rate, PRC Area—Precision-Recall Area, ROC Area—Receiver Operating Characteristic Area, MCC—Matthews Correlation Coefficient.

The classification models built based on a set of selected textures from images in color channel *L* were characterized by the lowest correctness (Figure 4). The overall accuracy reached 97.00% for the IBk algorithm. An accuracy of 100% was determined only for 'Leskovac' and a model built using IBk. The lowest overall accuracy of 94.00% was produced by a model built using Logistic.

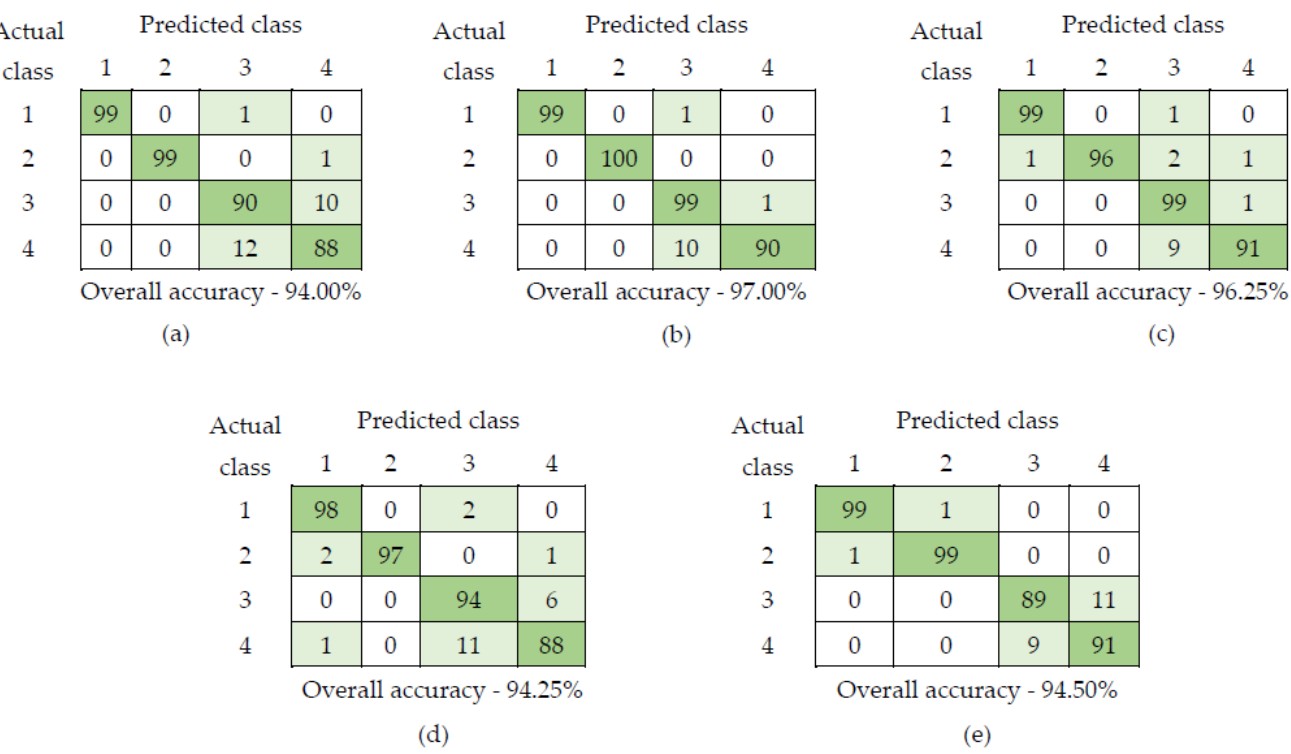

**Figure 4.** The confusion matrices of distinguishing quince seed cultivars 'Uspiech' (class 1), 'Leskovac' (class 2), 'Bereczki' (class 3), and 'Kaszczenko' (class 4) using models based on selected textures from images in color channel *L* developed using Logistic (**a**), IBk (**b**), LogitBoost (**c**), LMT (**d**), and naïve Bayes (**e**). Dark green color—correctly classified cases, light green color—incorrectly classified cases.

In the case of 'Leskovac' seeds and a model developed using the IBk algorithm, the performance metrics were the most successful. The TP Rate, Precision, PRC Area, ROC Area, MCC, and F-Measure were equal to 1.000, and FP Rate was 0.000 (Table 4). The values of Precision, PRC Area, ROC Area reaching 1.000, and FP Rate equal to 0.000 for

'Leskovac' were obtained for a model built using Logistic, and Precision of 1.000 and FP Rate of 0.000 were determined also for LogitBoost and LMT. Additionally, seeds of 'Uspiech' were classified with high correctness. The values of Precision, PRC Area, and ROC Area reached 1.000, and FP Rate was 0.000 in the case of Logistic, Precision of 1.000 and FP Rate of 0.000 were found for IBk, and ROC Area of 1.000 for naïve Bayes. The seed of 'Bereczki' and 'Kaszczenko' were classified with the highest FP Rate reaching 0.043 (Logistic, LMT) for 'Bereczki' and 0.037 (Logistic, naïve Bayes) for 'Kaszczenko'.

**Table 4.** The performance metrics of quince seed classification using models developed based on textures of images in color channel *L*.

| Algorithm | Class | TP Rate | FP Rate | Precision | PRC Area | ROC Area | MCC | F-Measure |
|---|---|---|---|---|---|---|---|---|
| Logistic (Functions) | 'Uspiech' | 0.990 | 0.000 | 1.000 | 1.000 | 1.000 | 0.993 | 0.995 |
| | 'Leskovac' | 0.990 | 0.000 | 1.000 | 1.000 | 1.000 | 0.993 | 0.995 |
| | 'Bereczki' | 0.900 | 0.043 | 0.874 | 0.935 | 0.984 | 0.848 | 0.887 |
| | 'Kaszczenko' | 0.880 | 0.037 | 0.889 | 0.963 | 0.984 | 0.846 | 0.884 |
| IBk (Lazy) | 'Uspiech' | 0.990 | 0.000 | 1.000 | 0.992 | 0.991 | 0.993 | 0.995 |
| | 'Leskovac' | 1.000 | 0.000 | 1.000 | 1.000 | 1.000 | 1.000 | 1.000 |
| | 'Bereczki' | 0.990 | 0.037 | 0.900 | 0.848 | 0.963 | 0.925 | 0.943 |
| | 'Kaszczenko' | 0.900 | 0.003 | 0.989 | 0.915 | 0.907 | 0.926 | 0.942 |
| LogitBoost (Meta) | 'Uspiech' | 0.990 | 0.003 | 0.990 | 0.998 | 0.999 | 0.987 | 0.990 |
| | 'Leskovac' | 0.960 | 0.000 | 1.000 | 0.998 | 0.999 | 0.973 | 0.980 |
| | 'Bereczki' | 0.990 | 0.040 | 0.892 | 0.980 | 0.994 | 0.919 | 0.938 |
| | 'Kaszczenko' | 0.910 | 0.007 | 0.978 | 0.988 | 0.995 | 0.926 | 0.943 |
| LMT (Trees) | 'Uspiech' | 0.980 | 0.010 | 0.970 | 0.971 | 0.996 | 0.967 | 0.975 |
| | 'Leskovac' | 0.970 | 0.000 | 1.000 | 0.986 | 0.987 | 0.980 | 0.985 |
| | 'Bereczki' | 0.940 | 0.043 | 0.879 | 0.966 | 0.989 | 0.877 | 0.908 |
| | 'Kaszczenko' | 0.880 | 0.023 | 0.926 | 0.960 | 0.982 | 0.872 | 0.903 |
| Naïve Bayes (Bayes) | 'Uspiech' | 0.990 | 0.003 | 0.990 | 0.999 | 1.000 | 0.987 | 0.990 |
| | 'Leskovac' | 0.990 | 0.003 | 0.990 | 0.993 | 0.991 | 0.987 | 0.990 |
| | 'Bereczki' | 0.890 | 0.030 | 0.908 | 0.914 | 0.980 | 0.866 | 0.899 |
| | 'Kaszczenko' | 0.910 | 0.037 | 0.892 | 0.948 | 0.980 | 0.868 | 0.901 |

TP Rate—True Positive Rate, FP Rate—False Positive Rate, PRC Area—Precision-Recall Area, ROC Area—Receiver Operating Characteristic Area, MCC—Matthews Correlation Coefficient.

The obtained results revealed the possibility of the classification of quince seeds belonging to cultivars: 'Uspiech', 'Leskovac', 'Bereczki', and 'Kaszczenko' with high overall accuracy reaching 98.75% for the Logistic from the group of Functions using selected image textures and traditional machine learning algorithms. The performed experiment was treated as supplementary to the previous studies of quince cultivar classification using models built based on texture parameters of flesh images, in which the total accuracies of 99% and 94% were obtained for images acquired using a digital camera and a flatbed scanner, respectively. The most successful algorithm was the Multilayer Perceptron also from Functions [30]. The present study confirmed that, in addition to the flesh, also the seeds can be a useful part of the quince fruit for cultivar classification. The usefulness of seeds for distinguishing cultivars was also revealed in previous own and other authors' research. For example, peach seeds belonging to two cultivars were correctly classified in 100% using models based on image textures developed using Bayes Net (Bayes), Logistic (Functions), SMO (Functions), and Multi Class Classifier (Meta) machine learning algorithms [31]. Models involving the image texture features of apple seeds classified two cultivars with an accuracy reaching 100% (naïve Bayes from the group of Bayes, Multilayer Perceptron from Functions and Multi Class Classifier) and three

cultivars in 85% (naïve Bayes) [32]. Sabanci et al. [33] distinguished four pepper seed cultivars in 99.02% using the SVM (Support Vector Machine) algorithm. Whereas Koklu et al. [34] correctly classified two types of pumpkin seeds in 88.64% using SVM.

The high classification accuracies obtained in this study are sufficient to distinguish quince seed cultivars with high probability. An overall accuracy of 98.75% determined for four classes allows for very correct classification. Such a result is sufficient to confirm the authenticity of seeds and to detect falsifications. However, it can prompt further research involving image analysis and machine learning for the quality evaluation of quince seeds. The appearance, including the color and structure of the seeds of different quince cultivars, may depend on the growing season and the degree of maturity of the fruit. Therefore, further more detailed studies will involve more cultivars, collected in several seasons and with different degrees of fruit maturity, to develop a more universal model for distinguishing quince seed cultivars. Another aspect of further research may concern the inclusion of geometric parameters such as linear dimensions and shape factors in classification models. Models combining the image texture parameters and geometric features could increase classification accuracy.

Additionally, the use of deep learning can result in a further increase in the cultivar classification accuracy of quince seeds. Deep learning is a modern approach to image processing and data analysis with great potential [13]. Traditional machine learning uses manually extracted features in image classification. It can be more time-consuming and complex and affects the classification performance. Therefore, computer vision often involves deep learning to extract image features [19]. Traditional classification models can be characterized by poor effects for the classification involving many categories and unbalanced sample distribution. Deep learning models can be successfully used to identify diverse seeds with high precision and the ability to deal better with a large amount of data. Deep learning can be distinguished from the traditional modeling pattern by adopting a structure, which is similar to the human brain [35]. Deep learning provides a hierarchical representation of data using various convolutions and thus can increase the learning capabilities and image classification performance metrics [36]. Therefore, performing research using deep learning can be beneficial.

## 4. Conclusions

The quince seeds belonging to four cultivars: 'Uspiech', 'Leskovac', 'Bereczki', and 'Kaszczenko' were the most successfully distinguished using the classification models based on attributes selected from sets of combined textures from images in all color channels *R*, *G*, *B*, *L*, *a*, *b*, *X*, *Y*, and *Z*. The optimal classification algorithm was Logistic from the group of Functions. The classification model built using Logistic based on a combined set of textures selected from images in all color channels *R*, *G*, *B*, *L*, *a*, *b*, *X*, *Y*, and *Z* was characterized by the highest overall accuracy of 98.75%. Despite the promising results of the cultivar classification of quince seeds using image processing and machine learning, there were some limitations and biases in the experimental data. The research was carried out in one growing season, for one degree of maturity and for a limited number of cultivars and seeds. Therefore, further research may be carried out using, for example, more quince cultivars and the next growing seasons. Additionally, with more seeds, deep learning models could be used.

**Author Contributions:** Conceptualization, E.R., D.K. and M.M.-F.; methodology, E.R.; software, E.R.; validation, E.R.; formal analysis, E.R.; investigation E.R.; resources, D.K.; data curation, E.R.; writing—original draft preparation, E.R., D.K. and M.M.-F.; writing—review and editing, E.R., D.K. and M.M.-F.; visualization, E.R.; project administration, D.K. All authors have read and agreed to the published version of the manuscript.

**Funding:** This work was performed in the frame of multiannual programme "Actions to improve the competitiveness and innovation in the horticultural sector with regard to quality and food safety

and environmental protection", financed by the Polish Ministry of Agriculture and Rural Development.

**Institutional Review Board Statement:** Not applicable.

**Data Availability Statement:** The data presented in this study are available upon request from the corresponding author.

**Conflicts of Interest:** The authors declare no conflict of interest.

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
