# Peer review of "Distinguishing Seed Cultivars of Quince (Cydonia oblonga Mill.) Using Models Based on Image Textures Built Using Traditional Machine Learning Algorithms"

_agriculture, doi:10.3390/agriculture13071310_

Round 1

Reviewer 1 Report

Insufficient rationality of experimental design: There are some problems in the experimental design of this paper. First of all, the characteristics of different varieties of quince seeds may be quite different, so it is necessary to select a suitable algorithm for classification. Secondly, the source of experimental data in this paper is limited, and there may be selection bias, which affects the reliability of experimental results. Therefore, it is necessary to improve the experimental design and improve its rationality.

Insufficient in-depth data analysis: There are some problems in the data analysis of this paper. Firstly, for different varieties of quince seeds, it is necessary to conduct in-depth analysis and comparison of their characteristics to find out the characteristics suitable for classification. Secondly, when analyzing the performance of the classification model, it is necessary to comprehensively consider various indicators, including accuracy rate, accuracy rate, recall rate, etc., in order to comprehensively evaluate the performance of the model. Therefore, it is necessary to conduct in-depth discussion and analysis on data analysis in order to improve the reliability and persuasiveness of the paper.

Conclusion reliability needs to be improved: There are some problems in the conclusion of this paper. Firstly, the classification performance of different varieties of quince seeds needs to be comprehensively compared and analyzed to find out the optimal classification algorithm. Secondly, when drawing a conclusion, it is necessary to consider the limitations and biases of experimental data to improve the reliability and universality of the conclusion. Therefore, it is necessary to improve and perfect the conclusion in order to improve its reliability and persuasiveness.

Author Response

Reviewer 1

Comment: Insufficient rationality of experimental design: There are some problems in the experimental design of this paper. First of all, the characteristics of different varieties of quince seeds may be quite different, so it is necessary to select a suitable algorithm for classification. Secondly, the source of experimental data in this paper is limited, and there may be selection bias, which affects the reliability of experimental results. Therefore, it is necessary to improve the experimental design and improve its rationality.

Response: The rationale for the studies performed is described in more detail as follows:

“The obtained results revealed the possibility of the classification of quince seeds belonging to cultivars: ‘Uspiech’, ‘Leskovac’, ‘Bereczki’, and ‘Kaszczenko’ with high overall accuracy reaching 98.75% for the Logistic from the group of Functions using selected image textures and traditional machine learning algorithms. The performed experiment was treated as supplementary to the previous studies of quince cultivar classification using models built based on texture parameters of flesh images, in which the total accuracies of 99% and 94% were obtained for images acquired using a digital camera and a flatbed scanner, respectively. The most successful algorithm was the Multilayer Perceptron also from Functions [30]. The present study confirmed that, in addition to the flesh, also the seeds can be a useful part of the quince fruit for cultivar classification.”

Therefore, the choice of algorithms resulted from previous experience.

The description of the experimental design has been supplemented with the following information:

“The combined textures of the images in each of the channels were useful for building the classification model. The textures with the highest power to distinguish quince seed cultivars were, among others, RSGNonZeros, RSGArea, GHMean, GHDomn10, BHMean, BHVariance, LHMean, LS5SN5SumAverg, aHMean, aHPerc99, bSGNonZeros, bS4RHGLevNonU, XS5SN1SumOfSqs, XS4RVGLevNonU, YS5SH1SumAverg, ZHMean, ZHMaxm10.”

Comment: Insufficient in-depth data analysis: There are some problems in the data analysis of this paper. Firstly, for different varieties of quince seeds, it is necessary to conduct in-depth analysis and comparison of their characteristics to find out the characteristics suitable for classification. Secondly, when analyzing the performance of the classification model, it is necessary to comprehensively consider various indicators, including accuracy rate, accuracy rate, recall rate, etc., in order to comprehensively evaluate the performance of the model. Therefore, it is necessary to conduct in-depth discussion and analysis on data analysis in order to improve the reliability and persuasiveness of the paper.

Response: The characteristics suitable for classification were indicated as follows:

“The textures with the highest power to distinguish quince seed cultivars were, among others, RSGNonZeros, RSGArea, GHMean, GHDomn10, BHMean, BHVariance, LHMean, LS5SN5SumAverg, aHMean, aHPerc99, bSGNonZeros, bS4RHGLevNonU, XS5SN1SumOfSqs, XS4RVGLevNonU, YS5SH1SumAverg, ZHMean, ZHMaxm10.”

“The differences in selected texture parameters, image texture between quin seed cultivars were determined at a significance level of p < 0.05 using STATISTICA 13.1 (Dell Inc., Tulsa, OK, USA, StatSoft Polska, Kraków, Poland). The normality of the distribution of variables and homogeneity of variance were checked. Then, Newman-Keuls test was used to compare the means.”

“The mean comparison of selected image textures is presented in Table 1.”

Table 1. The selected texture parameters of quince seeds.

Class

RSGNonZeros

GHMean

BHMean

LHMean

aHMean

ZHMean

ZHMaxm10

‘Uspiech’

0.992 a

74.22 a

63.27 a

116.32 a

142.47 a

15.78 a

0.678 a

‘Leskovac’

0.988 b

60.21 b

53.83 b

99.96 b

139.06 b

10.27 b

0.928 b

‘Bereczki’

0.987 c

45.55 c

39.86 c

84.36 c

137.82 c

6.07 c

0.956 c

‘Kaszczenko’

0.990 d

47.45 d

42.96 d

85.77 c

137.37 c

6.58 c

0.985 d

a,b - the same letters in the columns denote no statistical differences

The performance metrics of quince seed classification, such as TP (True Positive) Rate=Recall, FP (False Positive) Rate, Precision, PRC (Precision-Recall) Area, ROC (Receiver Operating Characteristic) Area, MCC (Matthews Correlation Coefficient), and F-Measure are presented in Tables 2-4 and discussed in the text in lines 169-215, 234-243, 261-271.

Comment: Conclusion reliability needs to be improved: There are some problems in the conclusion of this paper. Firstly, the classification performance of different varieties of quince seeds needs to be comprehensively compared and analyzed to find out the optimal classification algorithm. Secondly, when drawing a conclusion, it is necessary to consider the limitations and biases of experimental data to improve the reliability and universality of the conclusion. Therefore, it is necessary to improve and perfect the conclusion in order to improve its reliability and persuasiveness.

Response: Section 4. Conclusions has been improved as follows:

“The quince seeds belonging to four cultivars: ‘Uspiech’, ‘Leskovac’, ‘Bereczki’, and ‘Kaszczenko’ were the most successfully distinguished using the classification models based on attributes selected from sets of combined textures from images in all color channels R, G, B, L, a, b, X, Y, and Z. The optimal classification algorithm was Logistic from the group of Functions. The classification model built using Logistic based on a combined set of textures selected from images in all color channels R, G, B, L, a, b, X, Y, and Z was characterized by the highest overall accuracy of 98.75%. Despite the promising results of the cultivar classification of quince seeds using image processing and machine learning, there were some limitations and biases in the experimental data. The research was carried out in one growing season, for one degree of maturity and for a limited number of cultivars and seeds. Therefore, further research may be carried out using, for example, more quince cultivars and the next growing seasons. Additionally, with more seeds, deep learning models could be used.”

Reviewer 2 Report

 Distinguishing seed cultivars of quince (Cydonia oblonga Mill.) 2 using models based on image textures built using traditional 3 machine learning algorithms

The work presents the application of computer vision to identification of different cultivars of quince. The work is interesting, and may be relevant to the field of agricultural engineering and also image processing. The text is clear, and the methods are appropriate. However, I suggest the authors to improve the discussion of the results, to enhance the relevance of the work for the audience and future works on image processing or agricultural engineering, that may benefit from a better discussion about the results.

Introduction

The introduction clearly describes the importance of the work and establishes the background to justify its application. I suggest some minor changes regarding text format, as several sentences should be revised and merged, to provide a full meaning.

Material and methods

Figure 1: What is the relevance of showing channels G and L?

The figures of merit used are appropriate for model evaluation

Results and discussion

In my opinion, this is the main section that could and should be enhanced. The authors present a detailed text reporting what is already presented in Tables 1, 2 and 3, and Figures, 2, 3 and 4. After, in lines 296-321, the authors report several works that will be “further research involving image analysis and machine learning 299 for the quality evaluation of quince seeds. The appearance, including the color and struc-300 ture of the seeds of different quince cultivars, may depend on the growing season and the 301 degree of maturity of the fruit. Therefore, further more detailed studies will involve more 302 cultivars, collected in several seasons and with different degrees of fruit maturity, to de-303 velop a more universal model for distinguishing quince seed cultivars. Another aspect of 304 further research may concern the inclusion of geometric parameters such as linear dimen-305 sions and shape factors in classification models. Models combining the image texture pa-306 rameters and geometric features could increase classification accuracy. Additionally, the use of deep learning can result in a further increase in the cultivar 308 classification accuracy of quince seeds.”. All this description only involves extracting more parameters and applying different methods (deep learning). The authors should clarify why this was not already performed in the current experiment? Why not including further parameters from images and testing other methods such as deep learning?

I suggest the authors to fully reorganize this section, excluding or at least summarizing the text in lines 164-283, and discussing how these models reached such high accuracy. Why they were appropriate? Try to compare with other models for other types of seeds/grains, discussing if the current methods could be used for other samples, or if other methods from other works could also be appropriate or not, in the current study.

Also, the authors may try to explain why each model performed better or worse than others (although they provided quite similar figures of merit). Regarding the number of parameters (which is quite high, 1629), the authors could provide some insights about which of them were more important. I doubt all of them were useful, as probably 3-8 were mostly useful and the others had little relevance. I suggest to show the importance and discuss, as this may be useful for future researchers to use and compare the methods. Also, perhaps repeating channel colors from different color spaces may be redundant, perhaps only using one of the channels (RGB, Lab, etc) may already provide good results, and can be more useful for real applications, or also provide background for future works with different equipment. I believe this may greatly enhance the relevance of the work.

Minor comments

The text should be fully revised for typos and grammatical organization. Several sentences are not clear and should be rewritten. I hereby list a few of them, although there are several others:

Lines 34-35: short sentences, that could be merged to have a full meaning

Lines 43-44: Please merge sentences

Lines 54-57: Please merge this sentence with the sentence in lines 51-52

Please see comments to the authors

Author Response

Reviewer 2

Comment: The work presents the application of computer vision to identification of different cultivars of quince. The work is interesting, and may be relevant to the field of agricultural engineering and also image processing. The text is clear, and the methods are appropriate. However, I suggest the authors to improve the discussion of the results, to enhance the relevance of the work for the audience and future works on image processing or agricultural engineering, that may benefit from a better discussion about the results.

Response: Thank you very much for your careful reading of the manuscript. All your valuable comments have been taken into account. The discussion of the results has been improved.

Comment: Introduction

The introduction clearly describes the importance of the work and establishes the background to justify its application. I suggest some minor changes regarding text format, as several sentences should be revised and merged, to provide a full meaning.

Response: Some sentences have been revised and merged as you suggested. Please see the changed marked in red in the Introduction.

Comment: Material and methods

Figure 1: What is the relevance of showing channels G and L?

Response: It has been deleted. It was not necessary to show the images in channels G and L.

Comment: The figures of merit used are appropriate for model evaluation

Response: Thank you for your comment. We evaluated the model mainly on the accuracies shown in the Figures 2-4 and classification performance metrics presented in Tables 2-4.

Comment: Results and discussion

In my opinion, this is the main section that could and should be enhanced. The authors present a detailed text reporting what is already presented in Tables 1, 2 and 3, and Figures, 2, 3 and 4. After, in lines 296-321, the authors report several works that will be “further research involving image analysis and machine learning 299 for the quality evaluation of quince seeds. The appearance, including the color and

struc-300 ture of the seeds of different quince cultivars, may depend on the growing season and the 301 degree of maturity of the fruit.

Therefore, further more detailed studies will involve more 302 cultivars, collected in several seasons and with different degrees of fruit maturity, to de-303 velop a more universal model for distinguishing quince seed cultivars. Another aspect of 304 further research may concern the inclusion of geometric parameters such as linear dimen-305 sions and shape factors in classification models.

Models combining the image texture pa-306 rameters and geometric features could increase classification accuracy. Additionally, the use of deep learning can result in a further increase in the cultivar 308 classification accuracy of quince seeds.”. All this description only involves extracting more parameters and applying different methods (deep learning). The authors should clarify why this was not already performed in the current experiment? Why not including further parameters from images and testing other methods such as deep learning?

Response: The description of Tables and Figures has been limited. The discussion has been expanded as follows:

“The obtained results revealed the possibility of the classification of quince seeds belonging to cultivars: ‘Uspiech’, ‘Leskovac’, ‘Bereczki’, and ‘Kaszczenko’ with high overall accuracy reaching 98.75% for the Logistic from the group of Functions using selected image textures and traditional machine learning algorithms. The performed experiment was treated as supplementary to the previous studies of quince cultivar classification using models built based on texture parameters of flesh images, in which the total accuracies of 99% and 94% were obtained for images acquired using a digital camera and a flatbed scanner, respectively. The most successful algorithm was the Multilayer Perceptron also from Functions [30]. The present study confirmed that, in addition to the flesh, also the seeds can be a useful part of the quince fruit for cultivar classification. The usefulness of seeds for distinguishing cultivars was also revealed in previous own and other authors' research. For example, peach seeds belonging to two cultivars were correctly classified in 100% using models based on image textures developed using Bayes Net (Bayes), Logistic (Functions), SMO (Functions), and Multi Class Classifier (Meta) machine learning algorithms [31]. Models involving image texture features of apple seeds classified two cultivars with an accuracy reaching 100% (Naive Bayes from the group of Bayes, Multilayer Perceptron from Functions and Multi Class Classifier) and three cultivars in 85% (Naive Bayes) [32]. Sabanci et al. [33] distinguished four pepper seed cultivars in 99.02% using the SVM (Support Vector Machine) algorithm. Whereas Koklu et al. [34] correctly classified two types of pumpkin seeds in 88.64% using SVM.

The high classification accuracies obtained in this study are sufficient to distinguish quince seed cultivars with high probability. An overall accuracy of 98.75% determined for four classes allows for very correct classification. Such a result is sufficient to confirm the authenticity of seeds and to detect falsifications. However, it can prompt further research involving image analysis and machine learning for the quality evaluation of quince seeds.”

Comment: I suggest the authors to fully reorganize this section, excluding or at least summarizing the text in lines 164-283, and discussing how these models reached such high accuracy. Why they were appropriate? Try to compare with other models for other types of seeds/grains, discussing if the current methods could be used for other samples, or if other methods from other works could also be appropriate or not, in the current study.

Response: Section Results and discussion has been reorganized. The text in lines 164-283 has been limited. Whereas discussion has been expanded exactly as you suggested.

Comment: Also, the authors may try to explain why each model performed better or worse than others (although they provided quite similar figures of merit). Regarding the number of parameters (which is quite high, 1629), the authors could provide some insights about which of them were more important. I doubt all of them were useful, as probably 3-8 were mostly useful and the others had little relevance. I suggest to show the importance and discuss, as this may be useful for future researchers to use and compare the methods. Also, perhaps repeating channel colors from different color spaces may be redundant, perhaps only using one of the channels (RGB, Lab, etc) may already provide good results, and can be more useful for real applications, or also provide background for future works with different equipment. I believe this may greatly enhance the relevance of the work.

Response: The classification accuracy was the highest in the case of a model based on a combined set of textures selected from images in all color channels R, G, B, L, a, b, X, Y, and Z.

It has been corrected as follows:

“The textures of the images in each of the channels were useful for building the classification model. The textures with the highest power to distinguish quince seed cultivars were, among others, RSGNonZeros, RSGArea, GHMean, GHDomn10, BHMean, BHVariance, LHMean, LS5SN5SumAverg, aHMean, aHPerc99, bSGNonZeros, bS4RHGLevNonU, XS5SN1SumOfSqs, XS4RVGLevNonU, YS5SH1SumAverg, ZHMean, ZHMaxm10.”

Table 1 presenting the mean comparison of selected image textures has been added.

Table 1. The selected texture parameters of quince seeds.

Class

RSGNonZeros

GHMean

BHMean

LHMean

aHMean

ZHMean

ZHMaxm10

‘Uspiech’

0.992 a

74.22 a

63.27 a

116.32 a

142.47 a

15.78 a

0.678 a

‘Leskovac’

0.988 b

60.21 b

53.83 b

99.96 b

139.06 b

10.27 b

0.928 b

‘Bereczki’

0.987 c

45.55 c

39.86 c

84.36 c

137.82 c

6.07 c

0.956 c

‘Kaszczenko’

0.990 d

47.45 d

42.96 d

85.77 c

137.37 c

6.58 c

0.985 d

a,b - the same letters in the columns denote no statistical differences

Minor comments

Comment: The text should be fully revised for typos and grammatical organization.

Response: The text has been revised and corrected.

Comment: Several sentences are not clear and should be rewritten. I hereby list a few of them, although there are several others:

Lines 34-35: short sentences, that could be merged to have a full meaning

Lines 43-44: Please merge sentences

Lines 54-57: Please merge this sentence with the sentence in lines 51-52

Response: It has been corrected. Additionally, other improvements have been made.

Round 2

Reviewer 1 Report

The author has revised it as requested